# Tandem Mass Spectrometry as Strategy for the Selective Identification and Quantification of the Amyloid Precursor Protein Tyr682 Residue Phosphorylation Status in Human Blood Mononuclear Cells

**DOI:** 10.3390/biom11091297

**Published:** 2021-08-31

**Authors:** Pierluigi Reveglia, Rosarita Nasso, Antonella Angiolillo, Lucia Lecce, Carmela Paolillo, Samantha De Tullio, Monica Gelzo, Alfonso Di Costanzo, Carmela Matrone, Gaetano Corso

**Affiliations:** 1Department of Clinical and Experimental Medicine, University of Foggia, 71122 Foggia, Italy; pierluigi.reveglia@unifg.it (P.R.); lucia.lecce@unifg.it (L.L.); carmela.paolillo@unifg.it (C.P.); samanthadetullio75@gmail.com (S.D.T.); 2Division of Pharmacology, Department of Neuroscience, School of Medicine, University of Naples Federico II, 80131 Naples, Italy; rosarita.nasso@unina.it (R.N.); carmela.matrone@unina.it (C.M.); 3Centre for Research and Training in Medicine for Aging, Department of Medicine and Health Sciences “Vincenzo Tiberio”, University of Molise, 86100 Campobasso, Italy; angiolillo@unimol.it (A.A.); alfonso.dicostanzo@unimol.it (A.D.C.); 4CEINGE-Biotecnologie Avanzate, scarl, 80145 Naples, Italy; gelzo@ceinge.unina.it; 5Department of Molecular Medicine and Medical Biotechnology (DMMBM), University of Naples Federico II, 80131 Naples, Italy

**Keywords:** Alzheimer’s disease, APP Tyr682 phosphorylation, targeted peptide analysis

## Abstract

**Background:** Alzheimer’s disease (AD) is a devastating neurodegenerative disease without guidelines for early diagnosis or personalized treatment. Previous studies have highlighted a crucial role of increasing phosphorylation levels of the amyloid precursor protein (APP) Tyr682 residue in predicting neuronal deficits in AD patients. However, the lack of a method for the identification and quantification of Tyr682 phosphorylation levels prevents its potential clinical applications. **Methods:** Here we report a method to identify and quantify APP Tyr682 phosphorylation levels in blood mononuclear cells of AD patients by tandem mass spectrometry (tMS). **Results:** This method showed excellent sensitivity with detection and quantification limits set respectively at 0.035 and 0.082 ng injected for the phosphorylated peptide and at 0.02 and 0.215 ng injected for the non-phosphorylated peptide. The average levels of both peptides were quantified in transfected HELA cells (2.48 and 3.53 ng/μg of protein, respectively). Preliminary data on 3 AD patients showed quantifiable levels of phosphorylated peptide (0.10–0.15 ng/μg of protein) and below the LOQ level of non-phosphorylated peptide (0.13 ng/μg of protein). **Conclusion:** This method could allow the identification of patients with increased APP Tyr682 phosphorylation and allow early characterization of molecular changes prior to the appearance of clinical signs.

## 1. Introduction

Alzheimer’s disease (AD) is a progressive and irreversible neurodegenerative disorder, characterized by cognitive deficits and microscopic brain changes such as beta-amyloid plaques (Aβ plaques) outside neurons and twisted strands of the protein tau (tangles) inside neurons, and the latter two start being produced long before memory loss [1,2]. Unfortunately, despite the latest progresses in the knowledge of the mechanisms responsible for AD, the biggest challenge of the AD research still consists of finding new biomarkers for the early detection and the accurate diagnosis in the preclinical stages of AD. Indeed, defining appropriate criteria for a diagnosis in the very early stages is crucial, as early intervention is presumably the most effective.

Previous evidence has emphasized the role of the APP Tyr682 residue in processes responsible for the amyloid precursor protein (APP) trafficking and processing in neurons [3,4,5,6]. The APP Tyr682 residue is located in the highly conserved _682_YENPTY_687_ motif, which binds specific adaptor proteins depending on its phosphorylation state [5,7,8,9]. The APP interaction with these proteins acts as the major regulator of APP fate [5,6,7,10,11]. In particular, increased APP _Tyr682_ phosphorylation affects APP endocytosis and trafficking inside neurons [12]. Consequently, APP accumulates in the acidic neuronal compartments, such as late endosomes and lysosomes, where it is preferentially cleaved by β and γ secretases, to generate Aβ peptides [12,13]. We previously highlighted the critical role of APP Tyr682 phosphorylation in AD neurons, by promoting Aβ production and neuronal degeneration [14]. In fact, APP Tyr682 phosphorylation levels increase in familiar AD patients carrying mutations on the presenilin (PSEN) 1 gene as well as in sporadic AD patients [15,16,17]. In addition, Fyn tyrosine kinase (TK), which belongs to the Src TK family [18], elicits APP Tyr phosphorylation at level of the Tyr682 residue, and by doing this, triggers amyloidogenic processes in AD neurons [17]. Notably, the reduction of Fyn TK activity using either molecular or pharmacologic strategies, such as Fyn TK inhibitors (TKI: Saracatinib and Masitinib), prevents APP Tyr682 phosphorylation and Aβ accumulation and protects AD neurons from death [15,17]. Others have previously described a different involvement of Fyn in AD by increasing tau phosphorylation or triggering Aβ oligomer neurotoxicity mechanisms [19,20,21,22,23,24]. Recently, a Fyn upregulation in microglia cells of AD patients has been reported that precedes Aβ accumulation in neurons and contributes to neuroinflammation-associated synaptic dysfunction and neuronal damage [25,26].

Of note, we recently found that APP Tyr682 phosphorylation increases in fibroblasts of AD patients [16], opening up to the possibility to use APP Tyr682 phosphorylation level as a potential powerful tool in detecting early signs of AD-related cognitive disorders in peripheral cells. Indeed, cognitive performance reflects the integrity of the peripheral system and alterations in the peripheral system may compromise or exacerbate brain aging or brain dysfunctions [27]. In this context, having a diagnostic approach that allows the selective identification and quantification of APP Tyr682 phosphorylation status in peripheral cells of patients at risk of developing AD might help in the development of clinically relevant marker of disease onset and/or progression.

In this work we showed encouraging data supporting a targeting tandem mass spectrometry (tMS) approach to identify and quantify APP Tyr682 phosphorylation status in blood mononuclear cells of AD patients. We firstly settled and validated the procedure in HELA cell line in which APP and Fyn proteins were overexpressed and APP was phosphorylated on Tyr682 residue, as previously reported in neural stem cells [15]. Subsequently, we tested the procedure in blood mononuclear cells from AD patients. Our results, although limited to a restricted number of samples, prove that this procedure can be applied to blood mononuclear cells for identifying and quantifying APP Tyr682 phosphorylation levels.

## 2. Materials and Methods

### 2.1. Cell Lines

Human cell lines, HELA, were received from CEINGE Institute (Naples, Italy). Cells were grown in a humidified incubator at 37 °C under 5% CO_2_ atmosphere in MEM (Gibco, Rodano, MI, Italy) media supplemented with 10% FBS (Gibco, Rodano, MI, Italy), 2 mM of L-glutamine (Gibco, Rodano, MI, Italy), 100 IU/mL of penicillin G, and 100 μg/mL of streptomycin (Gibco, Rodano, MI, Italy). Cells were thawed every 2–3 weeks. Transfection experiments were always performed in low-passage cultures a maximum of 1 week after thawing, thus preserving the recovery of transfection activity. Cell transfections were usually carried out 24 h after plating.

### 2.2. Patients Selection and Isolation of Peripheral Blood Mononuclear Cells from Whole Blood

Patients with AD were diagnosed according to National Institute on Aging/Alzheimer’s Association (NIA/AA) criteria and fulfilled the criteria for “Probable AD dementia with evidence of the AD pathophysiological process” category. The severity of dementia was scored by Clinical Dementia Rating (CDR), which is estimated on the basis of patient’s and informant’s interview and physician’s clinical judgment [28,29]. The score is calculated based on assessing six cognitive and behavioral domains, including memory, orientation, judgment and problem solving, community affairs, home, and hobby performance, and personal care. The score ranged from zero to five as follows: no dementia (CDR = 0), very mild dementia (CDR = 0.5), mild dementia (CDR = 1), moderate dementia (CDR = 2), severe dementia (CDR = 3), very severe dementia (CDR = 4), and terminal dementia (CDR = 5). They had a Mini Mental State Examination (MMSE) score < 24, and positive PET amyloid imaging. To rule out other potential causes of cognitive impairment, patients underwent blood tests (including full blood count, erythrocyte sedimentation rate, urea and electrolytes, thyroid function, vitamin B12, and folate) and brain imaging. Peripheral blood mononuclear cells (PBMCs), including both lymphocytes and mononuclear cells, were separated from the whole blood by a density gradient centrifugation method using lymphocyte separation media Lymphosep (Biowest, Meda, MB, Italy). Blood samples were firstly diluted 1:1 with a saline phosphate buffer (PBS), and then carefully layered over Lymphosep in a 15-mL centrifuge tube, creating a sharp blood-Lymphosep interphase. Tubes were centrifuged at 400× *g* at room temperature for 30 min. The ring of mononucleate cells was collected and transferred into a new centrifuge tube. An equal volume of PBS was added to the PBMCs layer, and the mixture was centrifuged for 10 min at 260× *g* at room temperature. Cells were washed again with PBS to remove Lymphosep and platelets, and processed for cell lysate.

### 2.3. Transfection Experiments

EGFP-n1APP (Addgene: #69924) and pmApple-FYN-N-10 (Addgene: #54903) plasmids were purchased from Addgene (Teddington, UK). In HELA transient transfection experiments, 1 × 10^6^ cells were plated in a 25-cm^2^ flask. Twenty-four hours after plating, media were replaced with 3 mL of fresh medium and cells were transfected with a mix containing 5 µg of each DNA and 5 µL of Lipofectamine-2000 Transfection reagent (Thermofisher, Rodano, MI, Italy) in a final volume of 1 mL of pre-warmed Optimem medium (Gibco, Rodano, MI, Italy) in accordance with the manufacturer’s protocol. Transfected cells were incubated for 48 h at 37 °C. After 48 h transfection, cells were harvested, washed with PBS, and then lysed in ice-cold modified radio immunoprecipitation assay (RIPA) buffer (Sigma–Aldrich, Søborg, Denmark) supplemented with protease-phosphatase inhibitors (Merk Life Science S.r.l. Milan, Italy #A32959) and incubated for 30 min on ice. The supernatant obtained after centrifugation at 13,000 rpm for 20 min at 4 °C constituted the total protein extract.

### 2.4. Western Blot

In Western blot (WB) experiments, 30 µg of total protein HELA cells and 60 µg of total protein from blood mononuclear cell (PBMCs) of patients were loaded onto 8% polyacrylamide gel after being denatured at 95 °C for 5 min. Total protein concentration was determined by the Bradford method, using bovine serum albumin (BSA) as standard. For APP Tyr682 phosphorylation detection, 100 μg of total lysates from HELA cells was incubated overnight with phospho-Tyrosine Mouse mAb Magnetic Bead Conjugate (P-Tyr-100) (Cell Signaling, #8095; Milan, Italy). Immunoprecipitated (IP) samples were then analysed by WB. Gels were transferred to PVDF membranes (Immuno-Blot; Bio Rad, Segrate, MI, Italy) and incubated in primary rabbit anti APP antibody clone Y188 (Abcam, ab32136, Cambridge, UK) antibody and secondary monoclonal anti-β-actin-peroxidase (Merk Life Science S.r.l. MI, Italy) antibody.

### 2.5. SDS-PAGE and In-Gel Digestion

SDS–PAGE was performed using standard methods on the Bio-Rad Mini-Protean system, with 8% polyacrylamide gels. Then 30 µg of protein from HELA and 60 µg of protein from PBMCs of patients, both in reducing buffer, were loaded onto the gel after being denatured at 95 °C for 5 min. The gels were then run at 90–150 V for 1 h. Gels were stained with Coomassie Brilliant Blue (CBB) R250 and distained by shaking in 10% acetic acid in 30% methanol followed by rinsing in water. APP-GFP isoforms, migrating approximately from 100 to 150 kDa, were cut as a single band and divided into small cubes (1 mm) that were then collected into microcentrifuge tubes for the in-gel digestion following the protocol of Hellman protocol of Hellman et al., [30] slightly modified as follows. The gel pieces were dehydrated in 50 µL of acetonitrile for 15 min and rehydrated in 50 mM of AMBIC pH 8.0 (Merk Life Science S.r.l. MI, Italy) in incremental 5-min steps until CBB was removed. After 1 min spinning at 10,000 rpm, the supernatant was carefully discarded. The gel pieces were then rehydrated with 10 µL of trypsin solution (10 ng/µL, Promega; Italy) and incubated on ice for 60 min. After that, 50 mM of AMBIC was added to cover the gel pieces, and placed in a 37 °C water bath, overnight. After digestion, the hydrolysis mixture was centrifuged at maximum speed for a few seconds and the supernatant was collected in a new microcentrifuge tube. After that, 2% acetonitrile and 0.5% formic acid were added to each sample to stop enzymatic reaction. The remaining gel pieces were rinsed with 100 μL of acetonitrile for 15 min and the rinse was collected and combined with the previous supernatant. To dry supernatant down to dryness, samples were processed in speed vacuum.

### 2.6. Preparation of Samples and Standards

Three synthetic peptides: MQQNGYENPTYK, MQQNGpYENPTYK, and isotopically labelled MQQNGYENPTYK (Lys^13^C_6_,^15^N_2_), as internal standard, were obtained from GenScript Biotech (Piscataway, NJ, USA). Peptide standards solutions were prepared by diluting the 3 mg/mL of stock solution into water (0.1% formic acid) to reach the desired concentrations to build the calibration curve. Sample’s digested proteins were suspended in 90 µL of water (0.1% formic acid). All samples were spiked with 10 µL of I.S. at a final concentration of 2 µg/mL.

### 2.7. LC-MS/MS Analysis

We implemented the LC-MS/MS method by using a system consisting of an UPLC (Eksigent Ekspert ultraLC 100 series) coupled to a hybrid triple quadrupole/linear ion trap tandem mass spectrometer (QTRAP 4500, AB Sciex, Framingham, MA, USA) equipped with a Turbo V ion source. Instrument control, data acquisition, and processing were performed using the associated Analyst 1.6 software. The LC separation was carried out on a C18 column (EclipsePlus, 50 × 2.1 mm, RRHD particle size 3.5 µm) from Agilent (Santa Clara, CA, USA). Elution was performed at a flow rate of 300 µL/min with water containing 0.1% (*v*/*v*) formic acid as eluent A and ACN (Merck, Darmstadt, Germany) containing 0.1% (*v*/*v*) as eluent B, employing a linear gradient from 100% A to 50% A in 6 min. The injection duty cycle was 11 min, considering the column equilibration time. Q1 resolution was adjusted to 0.7 ± 0.1 amu fwhm for MRM, referred to as the unit resolution. Q3 was also set to the unit resolution in MRM mode. MS analysis was carried out in positive ionization mode using an ion spray voltage of 5000 V. The nebulizer and the curtain gas flows were set at 60 psi using nitrogen. The Turbo V ion source was operated at 400 °C with the auxiliary gas flow (nitrogen) set at 50 psi. Two suitable MRM transitions were selected for the peptides MQQNGYENPTYK and MQQNGpYENPTYK, while one transition was selected for the IS MQQNGYENPTYK(Lys^13^C_6_,^15^N_2_). The compounds dependent parameters for the three synthetic peptides were optimized using the manual optimization protocol in tuning mode. The Q1 mass, the Q3 transition, and the best parameters are reported in Table 1.

### 2.8. Information-Dependent Acquisition (IDA) Parameters

An IDA experiment was performed to automatically trigger EPI scans by analyzing MRM signals. The IDA criteria were set to select one intense peak exceeding 500 counts/s and without exclusion after dynamic background subtraction of the survey scan. The mass tolerance was set at 250 mDa. For the EPI scan, the scan rate was set to 1000 Da/s from 100 to 1000 Da. The CE was set at 40 eV, and the collision energy speed (CES) of the EPI was set at 15 eV to provide rich EPI spectra. MS/MS spectra of unknown samples were compared to standard spectra.

### 2.9. Validation

A validation study was obtained analyzing calibration curves, limit of detection (LOD), limit of quantification (LOQ), and within-day and between-day imprecision and inaccuracy. Calibration curves were obtained reporting the area ratio (peptide area/I.S. area) against the injected ng of peptide. Five microliters of each peptide standard solution, corresponding to 0.25, 2.5, 5, 10, 25, and 45 ng, were injected in triplicate to build the calibration curves. Two different curves were built for the MQQNGYENPTYK and MQQNGYpENPTYK peptides, respectively. LOD and LOQ were calculated as a magnitude of, respectively 3 and 10 times the standard deviation of noise to the lower point of standard level (0.25 ng). Two quality control samples (QC) at two different concentration levels were used for assessing the within-day and between-day variation. The lowest QC level consisted of an injection of 3.75 ng of the peptides, the highest QC level consisted of an injection of 35 ng of the peptides. The within-day imprecision (CV%) and inaccuracy (%) were calculated analyzing, in the same analytical run, each level of QC samples 3 times. The between-day imprecision (CV%) and inaccuracy (%) were calculated analyzing each level of QC samples once a day for 5 days.

### 2.10. Statistical Analysis and Data Processing

Data were processed and analyzed by MultiQuant Software version 3.0.2. Three independent experiments in HELA cell were analyzed in triplicate. The AD patients’ samples were analyzed in triplicate (technical replicate). Data are expressed as mean ± standard deviation of the mean.

## 3. Results

### 3.1. Tandem Mass Spectrometry Set Up and Validation

The MRM-EPI method consists of a short analytical cycle of 6 min and requires minimum sample preparation for the LC-MS/MS analysis. Figure 1 shows the extract ion chromatogram (XIC) and EPI spectrum of standard MQQNGYENPTYK peptide, while Figure 2 reports the XIC and EPI of standard MQQNGYpENPTYK peptide. The doubly-protonated forms, [M + 2H]^2+^, of the standard peptides were the most abounded during the optimization process. Thus, they were selected as precursor ions for the optimization of the MRM-EPI experiments.

The validation was obtained by analyzing calibration curves and quality controls. The linearity of calibration curves for both peptides, MQQNGYENPTYK and MQQNGYpENPTYK, was estimated by the coefficients of correlation (r) which were respectively 0.9972 and 0.9984. The slopes were 0.30970 and 1.05976 and intercept values were −0.00604 to −0.00309 (Table 2).

Quality control samples were analyzed after a sequence of unknown samples. Two levels of QC samples were prepared to evaluate imprecision and inaccuracy of the method. The within-day imprecision (CV%) and inaccuracy (%) were calculated by analyzing, in the same analytical run, each level of QC samples for 6 times. The imprecision varied between 3.2% to 5.2% for MQQNGYENPTYK and 2.1% to 4.9% for MQQNGYpENPTYK. The inaccuracy (%) ranged from 4.6% to 11.3% for MQQNGYENPTYK, and from −1.4% to 7.7% for MQQNGYpENPTYK (Table 3).

The between-day imprecision (CV%) and inaccuracy (%) were calculated analyzing each level of QC samples once a day for 5 days. The imprecision was estimated from (CV%) 6.6% to 8.6% for MQQNGYENPTYK and from 9% to 14.3% for MQQNGYpENPTYK; the inaccuracy was estimated from (%) 3.8% to 9.2% for MQQNGYENPTYK and 2.3% for MQQNGYpENPTYK. (Table 4).

### 3.2. Samples Analysis

HELA cells were transfected with C-terminal green fluorescent protein (GFP) tagged APP and C-terminal apple tagged Fyn. The efficiency of the transfection was evaluated by WB using rabbit anti-APP and rabbit anti-Fyn antibodies 48 h after transfection (Appendix A). WB analysis of total lysates showed the presence of bands at approximately 150 and 85 kDa corresponding to the molecular weight (MW) of the APP- and Fyn-tagged fused proteins, respectively, thus indicating that cells were properly transfected (Appendix A). In order to confirm that APP and Fyn overexpression results in the increased APP Tyr682 phosphorylation, samples were immunoprecipitated with anti-phospho-Tyr magnetic conjugated beads and analyzed by WB using rabbit anti-APP antibody. As shown in Appendix A, APP was phosphorylated on Tyr residue in APP+Fyn co-transfected cells.

Total lysate from APP and Fyn co-transfected cells were loaded on SDS-Tris-Glycine gel. The bands migrating between 100 and 150 kDa, including the three different isoforms of APP were cut and digested following the procedure above-described. The samples after proteolysis were processed by LC-MS/MS. The explorative analysis of the extract from transfected HELA cells showed the presence of both MQQNGYpENPTYK and MQQNGYENPTYK peptides, in a variable amounts (ng/μg of protein) of 3.53 ± 0.07 and 2.48 ± 0.26, respectively (Figure 3A).

These results encouraged the possibility to apply the procedure to analyze cell lines and blood cells, and it was sensitive enough to detect and quantify changes in the APP Tyr682 phosphorylation.

Therefore, we experimented using this method on mononuclear blood cells collected from three AD patients (Table 5). The severity of dementia of the these patients was scored by Clinical Dementia Rating (CDR), which is estimated on the basis of patient’s and informant’s interview and physician’s clinical judgment [28,29].

We isolated and purified blood mononuclear cells as reported in the Methods, and we processed samples as previously described for HELA cells. Figure 3B reports an example of MRM ion chromatograms from blood mononuclear cells of one of the three AD patients analyzed. tMS showed a difference in the levels of MQQNGYENPTYK and MQQNGYpENPTYK. In particular, the phosphorylated peptide (MQQNGYpENPTYK) was present in the mononuclear cells of the AD patients, resulting in 0.10 ± 0.03 ng/μg of protein for G.A. (Appendix A) and 0.15 ± 0.09 ng/μg of protein for O.S. (Appendix A). On the other hand, the unphosphorylated MQQNGYENPTYK peptide was detectable only in O.S., in an amount of 0.13 ± 0.002 ng/μg of protein (Appendix A). In the G.A. sample, MQQNGYENPTYK was below the LOD, although distinguishable from background noise (data not shown). In the T.V. sample, both peptides were below the LOD (data not shown).

## 4. Discussion

Previous evidence supports the hypothesis that changes in the APP Tyr682 phosphorylation status influences APP trafficking and promotes the amyloidogenic APP processing to generate Aβ in neurons [5,7,11,12,13,14,15,17,31]. Notably, we recently suggested that changes related to APP Tyr682 phosphorylation in fibroblasts may reflect Aβ-related abnormalities in the brain [16], thus emphasizing the promising potential of using APP Tyr682 phosphorylation levels to develop new diagnostic strategies and to optimize therapeutic approaches with better outcomes in patients who are included in clinical trials [5,6,16]. Indeed, all these promising results need the development of a selective and quantitative procedure for the APP Tyr682 phosphorylation detection. In this regard, it is worth mentioning that APP has three Tyr(s) along the C-terminal tail [5], and although Tyr phosphorylation at sites different than Tyr682 seems to not affect the extent of A*β* production or accumulation in neurons, the possibility that changes in the phosphorylation states of the other Tyr(s) can still interfere with neuronal functions cannot be ruled out. Thus, it is necessary to develop a procedure that can discriminate among the three Tyr(s) and selectively assess the levels of Tyr682 phosphorylation.

Herein, we described for the first time a fast and accurate LC-MS/MS method for simultaneous quantification of non-phosphorylated-MQQNGYENPTYK and phosphorylated-MQQNGYpENPTYK peptides coming from tryptic digestion of the APP containing the Tyr682 residue. Data from calibration curves and quality controls showed that the MRM analysis of peptides by LC-MS/MS method has good specificity, accuracy, and precision, and it can be used for its application in any type of cultured, tissue- or blood-isolated cells that expressed APP. Moreover, we provided evidence that this procedure is sensitive enough to detect APP Tyr682 phosphorylation levels injecting only 5 μL of the sample. However, the real strength of the method is its high specificity in detecting explicitly and exclusively APP-Tyr682 residue phosphorylation. The application of this methods on mononuclear cells from blood of patients with AD can definitely proof whether and in which level the increase in APP Tyr phosphorylation previously demonstrated by immunoprecipitation [16] is correlated to the specific APP Tyr682 residue. Even though the chromatographic peaks are very close and appear to be not completely separated, MS/MS technology enables to distinguish MQQNGYENPTYK and MQQNGYpENPTYK unambiguously by their different fragmentation pattern as demonstrated by EPI-MRM experiments. Mass spectrometry has been widely used to study the phosphorylation of proteins involved in neurodegeneration deposition within areas of the cerebral cortex, basal ganglia, and/or spinal cord, whilst the association between clinical phenotype and protein dysfunction have not been completely clarified so far [32,33]. Usually, untargeted approaches and a high-resolution platform are used [34,35,36,37]. However, currently these instrumentations have no application in clinical laboratory. On the contrary, we developed a targeted MS method to quantify a specific phosphorylation of APP using a low-resolution platform that is more and more frequently being used for clinical application [38].

Of note, this is the first time that the MQQNGYpENPTYK has been detected in the blood mononuclear cells of patients with AD, placing the basis for further analysis to verify the potential role of APP Tyr682 phosphorylation as biomarker in AD.

Moreover, this approach is also useful to calculate the ratio between phosphorylated and unphosphorylated peptides.

## 5. Conclusions

Collectively, these findings highlight the utility of this novel tMS approach to selectively identify and analyze APP Tyr682 phosphorylation levels in human mononuclear cells, as well as other cells. Although this procedure has some limitations that still require further improvement, such as the inclusion of more patients and healthy individuals. Indeed, it will be necessary to define the reference values in the blood, and the ratio between phosphorylated and non-phosphorylated peptides. The latter could represent the indicator of the level of APP-Tyr682 phosphorylation which may assume a critical clinical significance in patients with AD. In this regard, and in line with our previous findings, our analysis of APP Tyr682 phosphorylation levels in blood mononuclear cells might be used as biomarker for early diagnosis or for monitoring AD progression. In addition, it might allow the stratification of patients before being included in clinical trials and the follow-up assessments in the efficacy trials.

## Figures and Tables

**Figure 1 biomolecules-11-01297-f001:**
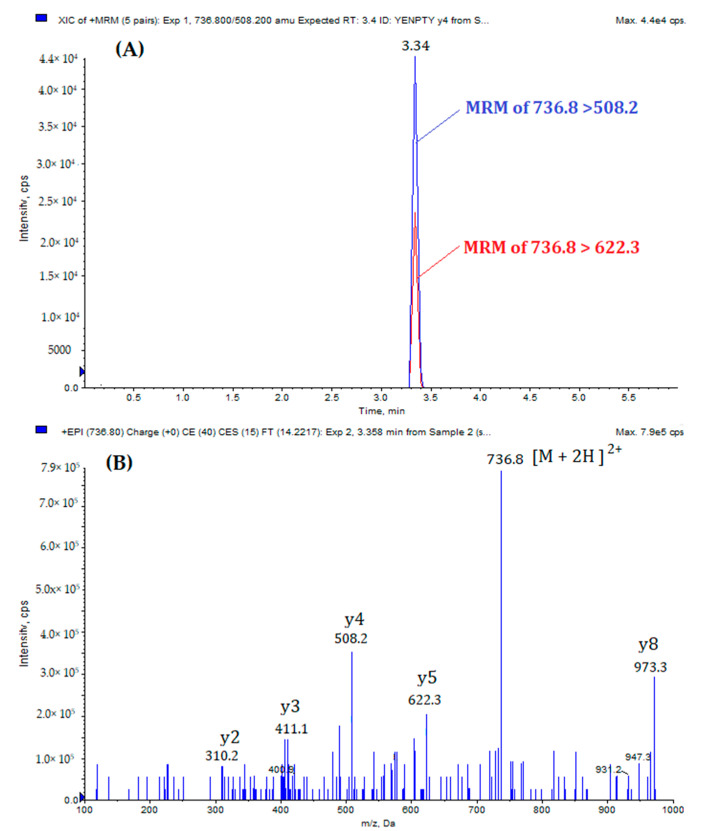
(**A**) Extract ion chromatogram of MQQNGYENPTYK: MRM transition 736.8 > 508.2 (blue) and 736.8 > 622.3 (red). (**B**) EPI spectrum of MQQNGYENPTYK.

**Figure 2 biomolecules-11-01297-f002:**
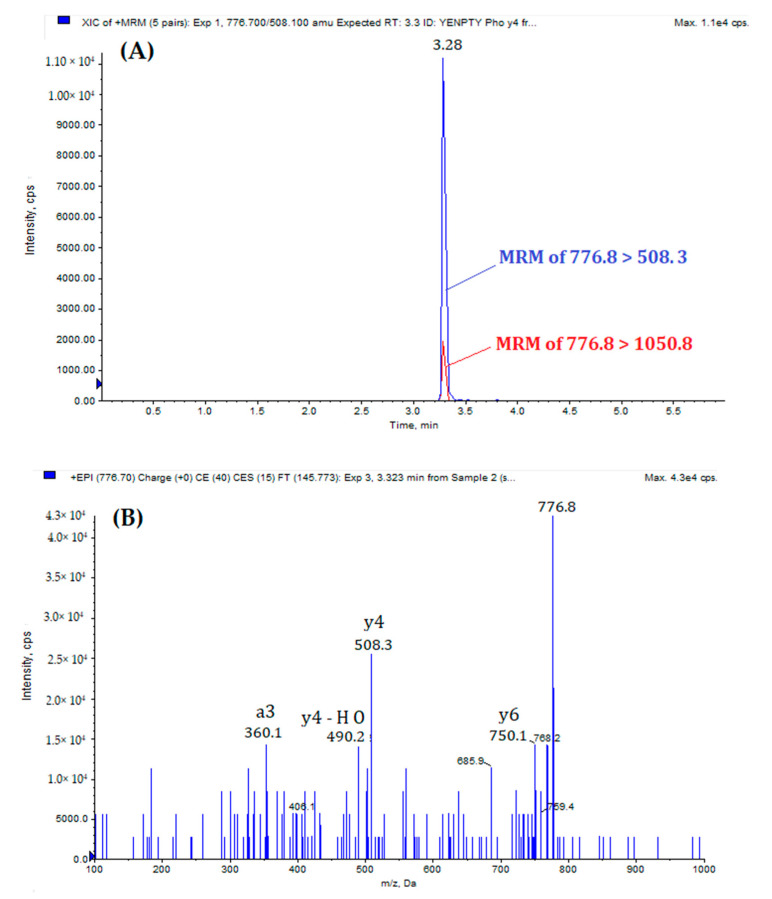
(**A**) Extract ion chromatogram of MQQNGYpENPTYK: MRM transition 776.8 > 508.3 (blue) and 776.8 > 1050.8 (red). (**B**) EPI spectrum of MQQNGYpENPTYK.

**Figure 3 biomolecules-11-01297-f003:**
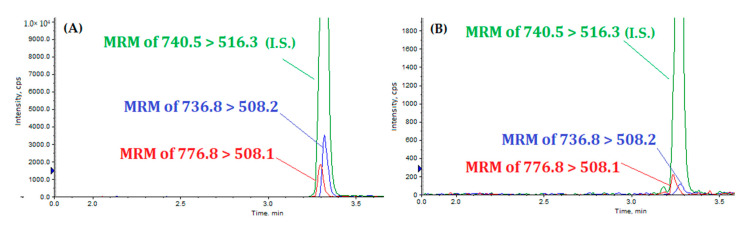
(**A**) Extract ions chromatograms of MQQNGYENPTYK(Lys13C6,15N2) (green:I.S.), MQQNGYpENPTYK (red), and MQQNGYENPTYK (blue) in HELA cells; (**B**) extract ions chromatograms of MQQNGYENPTYK(Lys13C6,15N2) (green), MQQNGYpENPTYK (red), and MQQNGYENPTYK (blue) in AD patient.

**Table 1 biomolecules-11-01297-t001:** Optimized Q1 mass, transitions, and parameters for MRM experiment.

Peptide		Precursor Ion (*m*/*z*)	Product Ion (*m*/*z*)	DP ^a^	CE ^b^	CXP ^c^	RT ^d^ (min)
**MQQNGYENPTYK**	(**Quantification**)	736.8 [M +2H]^2+^	508.2	55	41	25	3.34
**(Qualification)**		736.8	622.3	55	37	25	
**MQQNGYpENPTYK**	(**Quantification**)	776.7 [M +2H]^2+^	508.1	55	41	25	3.28
**(Qualification)**		776.7	1050.8	55	37	25	
**MQQNGYENPTYK(Lys^13^C_6_,^15^N_2_)**		740.5 [M +2H]^2+^	516.3	55	41	25	3.34

^a^ DP = declustering potential; ^b^ CE = collision energy; ^c^ CXP = collision cell exit potential; ^d^ RT = retention time.

**Table 2 biomolecules-11-01297-t002:** Calibration curves, correlation coefficient (r), imprecision, inaccuracy, and LOD and LOQ values for MQQNGYENPTYK and MQQNGpYENPTYK quantification.

Peptide	LOD	LOQ	ng of Peptide Injected	0.25	2.5	5	10	25	45
**MQQNGYENPTYK** ***y* = 1.05976 × −0.00604** ***r* = 0.9984**	0.02	0.215	Mean	0.27	2.41	5.10	9.49	23.61	46.88
			Imprecision ^a^	6.1	7.4	2.0	1.5	5.3	2.5
			Inaccuracy ^b^	8.0	−3.6	2.0	−5.0	−5.6	4.1
**MQQNGYpENPTYK** ***y* = 0.30900 × −0.00187** ***r* = 0.9972**	0.035	0.082	Mean	0.30	2.48	4.66	8.85	23.49	47.49
			Imprecision ^a^	16.6	7.6	2.3	2.5	5.4	2.2
			Inaccuracy ^b^	20	−0.8	−6.2	−11.5	−6.2	4.4

^a^ Expressed as relative standard deviation (CV%): (standard deviation/mean) × 100. ^b^ Expressed as % difference: [(concentration observed − concentration added)/concentration added] × 100.

**Table 3 biomolecules-11-01297-t003:** Within-day imprecision (CV%) and inaccuracy (%) of results obtained by measuring two QC levels 6 times within one day (*n* = 6).

Peptide	Expected Amount (ng)	Calculated Amount (ng)	Imprecision (CV%)	Inaccuracy (%)
**MQQNGYENPTYK**	3.75	4.15	5.2	11.3
	35.0	36.6	3.2	4.6
**MQQNGYpENPTYK**	3.75	3.7	2.1	−1.4
	35.0	37.7	4.9	7.7

**Table 4 biomolecules-11-01297-t004:** Between-day imprecision (CV%) and inaccuracy (%) of results obtained by measuring two QC levels once a day for five days (*n* = 5).

Peptide	Expected Amount(ng)	Calculated Amount (ng)	Imprecision (CV%)	Inaccuracy (%)
**MQQNGYENPTYK**	3.75	4.1	6.6	9.2
	35.0	36.35	8.6	3.8
**MQQNGYpENPTYK**	3.75	3.75	14.3	-
	35.0	35.8	9.0	2.3

**Table 5 biomolecules-11-01297-t005:** Detailed information of patients studied and amounts of peptides extracted after the in-gel digestion of APP.

Patients I.D.	Age (y)	Gender	A.D. Specific Mutation	Stage of Dementia (CDR) *	Peptide Tyr682 Phosphorylated (ng/μg of Prot)	Peptide No-Tyr682 Phosphorylated (ng/μg of Prot)
**O.S.**	78	M	None	Mild (1)	0.15	0.13
**G.A.**	80	F	None	Severe (3)	0.10	<LOD
**T.V.**	75	M	None	Terminal (5)	<LOD	<LOD

* CDR: Clinical Dementia Rating by ref. Hughes et al. 1982; Heyman et al. 1987 [28,29].

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
