# Peer review of "Tandem Mass Spectrometry as Strategy for the Selective Identification and Quantification of the Amyloid Precursor Protein Tyr682 Residue Phosphorylation Status in Human Blood Mononuclear Cells"

_biomolecules, 2021, doi:10.3390/biom11091297_

Round 1

Reviewer 1 Report

This study applies tandem mass spectrometry to measure the phosphorylation levels of Tyr682 in human blood mononuclear cells. This is an important study for development of AD biomarkers with potentially important applications. The data is interesting, but  too preliminary. The manuscript will be suitable for publication after the authors address the issues below.

  1. Using as a control the phosphorylation levels of APP transfected in HeLa cells is not a sufficient, as Hela cells are quite different than human blood cells. The authors need to include in the manuscript measurements of Tyr682 phosphorylation levels in samples acquired from blood samples of aged-matched and gender-matched individuals that are healthy.
  2. The number of blood samples is too low. The conclusions will be more convincing if the authors show data from 7-8 different blood samples of patients with AD.
  3. For each individual/patient, the authors need to include a table with the following details:

age of the patient

genetics (any AD-specific mutations)

gender of the patient

stage of disease

  1. No phosphatase inhibitors were used, raising the possibility that some phosphorylation might had occurred while the blood samples were being processed. Thus the authors need to include the following data
  • time for processing the blood samples.
  • repeat all measurements with samples to which phosphatase inhibitors were added
  1. Similarly no phosphatase inhibitors were used when processing the transfected HeLa cells with EGFP-n1APP and pmApple-FYN-N-10 plasmids. Thus the authors need to repeat the experiments using phosphatase inhibitors.
  2. In regard to statistical data analysis, it is not clear what ‘triplicates’ mean. Do the authors refer to the triplicates that are acquired by repeated measurements of the sample (technical replicates)? OR the triplicates refer to data from independent experiments (biological replicates) in which samples were acquired from different cell preparations of HeLa and the blood samples that were drown in at least three different times from the patients/controls? The authors need to clarify this. For the data to be published the authors need to show data from three biological replicates.
  3. In Figures 1 and 2 what does the blue and red color refer to? The authors need to include an explanation of the colors in the figure caption. Additionally the authors need to explain the top and the bottom spectrum and label them accordingly.
  4. The  mass spec data of the blood samples of all three patients needs to be shown.

Reviewer 2 Report

The study identified and quantified APP Tyr682 phosphorylation levels in blood mononuclear cells of AD patients. However, a few minor issues need to be addressed:

Page 1; Line 29-30: Abstract needs to be constructed in a way that all the outcomes with the significant figures should be given either in the percentage or statistically significant values.

Page 3; Line 89: What was the passage of the cells?

Page 3; Line 93: Is 24 hrs enough for HELA to reach 80% of the confluency?

Page 4; Line 169: Is LC-MS/MS analysis method has been developed or adapted from elsewhere?

Round 2

Reviewer 1 Report

Data of blood samples of healthy individuals is still lacking. Without this data it is really hard to judge whether the method can efficiently detect the change of the phosphorylation levels of APP due to AD. 

In the discussion , the authors need to elaborate more in detail on how their LC-MS/MS method compares with other MS methods used for detecting phosphorylation of proteins. Clear explanation of the possible advantages and limitations of their method need to be included.
